# Low-Cost Non-Wearable Fall Detection System Implemented on a Single Board Computer for People in Need of Care

**DOI:** 10.3390/s24175592

**Published:** 2024-08-29

**Authors:** Vanessa Vargas, Pablo Ramos, Edwin A. Orbe, Mireya Zapata, Kevin Valencia-Aragón

**Affiliations:** 1Grupo de Investigación Embsys, Departamento de Eléctrica, Electrónica y Telecomunicaciones, Universidad de las Fuerzas Armadas ESPE, Av. General Rumiñahui y Ambato, Sangolquí 171103, Ecuador; pframos@espe.edu.ec; 2Grupo de Investigación Embsys, Carrera de Ingeniería en Electrónica y Automatización, Universidad de las Fuerzas Armadas ESPE, Av. General Rumiñahui y Ambato, Sangolquí 171103, Ecuador; eaorbe@espe.edu.ec; 3Centro de Investigación en Mecatrónica y Sistemas Interactivos (MIST), Ingeniería Industrial, Universidad Indoamérica, Av. Machala y Sabanilla, Quito 170103, Ecuador; mireyazapata@uti.edu.ec (M.Z.); kg.va1234@gmail.com (K.V.-A.)

**Keywords:** fall-detection, non-wearable, deep-learning, artificial vision, CNN, single board computer, people in need of care

## Abstract

This work aims at proposing an affordable, non-wearable system to detect falls of people in need of care. The proposal uses artificial vision based on deep learning techniques implemented on a Raspberry Pi4 4GB RAM with a High-Definition IR-CUT camera. The CNN architecture classifies detected people into five classes: fallen, crouching, sitting, standing, and lying down. When a fall is detected, the system sends an alert notification to mobile devices through the Telegram instant messaging platform. The system was evaluated considering real daily indoor activities under different conditions: outfit, lightning, and distance from camera. Results show a good trade-off between performance and cost of the system. Obtained performance metrics are: precision of 96.4%, specificity of 96.6%, accuracy of 94.8%, and sensitivity of 93.1%. Regarding privacy concerns, even though this system uses a camera, the video is not recorded or monitored by anyone, and pictures are only sent in case of fall detection. This work can contribute to reducing the fatal consequences of falls in people in need of care by providing them with prompt attention. Such a low-cost solution would be desirable, particularly in developing countries with limited or no medical alert systems and few resources.

## 1. Introduction

According to the World Health Organization, falls constitute the second leading cause of unintentional injury death worldwide, after road traffic injuries. Annually, an estimated 684,000 people die due to accidental falls, often resulting from unintentional tripping or slipping. Notably, one in three adults over 65 experiences a fall each year [1,2]. Globally, the prevalence of falls among people over 60 is 26.5%, with Oceania having the highest rate by continent with 34.4%, followed by America (27.9%), Asia (25.8%), Africa (25.4%), and Europe (23.4%) [3].

Although falls can affect everyone, the severity of the consequences is influenced by factors such as age, health status, and the response time to assist the emergency. For elderly individuals, falls often result in severe consequences such as bruises, hip fractures, and traumatic brain injuries. It is well-documented that these incidents significantly diminish the quality of life for this age group [4,5]. This situation is aggravated because they do not receive prompt attention due to a lack of warning since they are often alone. Consequently, addressing elderly falls is a public concern that requires support from families and healthcare systems. Although medical alert systems with fall detection are available for a monthly fee in developed countries, accessing these services in developing countries remains a major challenge. This underscores the need for a cost-effective solution that can monitor falls and automatically notify family members, ensuring timely medical attention.

In general, two main techniques are used for fall detection: machine learning and sensor-based approaches [6]. Several studies on fall detection are based on wearable technology to monitor and alert from falls [7]. Recently, these systems have integrated artificial intelligence algorithms, encompassing both machine learning (ML) and deep learning (DL) [8,9]. Some implementations employ cameras to facilitate artificial vision algorithms, while others incorporate sensors like accelerometers, gyroscopes, and air pressure sensors for fall detection [10,11,12]. Other works deal with portable devices with remote monitoring systems [13,14]. However, the effectiveness of these wearable devices relies heavily on their proper use, as they are typically worn as clothing or accessories [15]. Consequently, the use of wearable devices may not be the most practical option for seniors who often forget or resist using additional accessories. Additionally, a relevant drawback is their reliance on batteries for operation. Nevertheless, compared to devices that require the use of cameras, wearable devices offer a more discreet monitoring solution, which can be perceived as less intrusive.

On the other hand, non-wearable systems are designed to be installed in the person’s environment [16]. These systems typically utilize artificial vision technologies, employing cameras, infrared sensors, and laser range scanners, among others [17,18]. Artificial vision systems capture and process images to determine if a fall has occurred. Non-wearable systems offer some advantages, particularly for the elderly, as they spend much of their time in a fixed location. Also, continuous monitoring identifies and mitigates potential risks in people’s environments. Moreover, its functionality does not depend on the user’s activities, and its cost is higher compared to wearable devices.

This paper introduces a fall detection system based on artificial intelligence implemented on a low-cost single board computer (SBC). This system was primarily designed for indoor nursing homes. The main contribution of this work is the implementation of a cost-effective and easy to install non-wearable fall detection system which send an alert notification via the Telegram application when a fall is detected. Early notifications of such events reduce the risk of severe injuries. Tests were conducted with persons going about daily activities, achieving a good trade-off between cost and performance. The remainder of this article is organized as follows: Section 2 presents related work, Section 3 details the materials and methods, Section 4 presents the results of different tests, Section 5 discusses the obtained results, and finally, Section 6 concludes this article.

## 2. Related Work

This section presents a summary of relevant works regarding non-wearable fall detection systems. The study conducted by Francy Shu and Jeff Shu [19] presented a sophisticated system capable of detecting falls, trips, slips, blackouts, and various types of falls. This system is based on a conventional Android TV Box (Manufactered by Tanix in Shenzhen, China) with 8 IP cameras and an eight-core S912 cortex-A53 CPU (By Amlogic, Shnaghai, China) for processing. This work uses Relevance Vector Machine (RVM) algorithms and AI algorithms offered by SpeedyAI Inc. (Walnut, CA, USA), achieving an 89% retention accuracy, and a training accuracy of 94%.

Ref. [20] presented a Convolutional Neural Network (CNN)-based fall detection system that addresses issues such as precise recognition of small and obscured body parts. It is developed with a lightweight feature extraction model, global and local attention modules, and channel-based feature augmentation. It focuses on medical, therapeutic, and clinical contexts.

The authors of [21] used WiFi Channel State Information (CSI) for fall detection using deep learning techniques. The construction of a comprehensive dataset for fall detection using CSI data are detailed. The paper provides valuable insights into the challenges and potential solutions for WiFi CSI-based fall detection systems. Also, the authors emphasize the need for further research to improve the applicability of fall detection systems in real-world scenarios.

The work presented in [1] introduced a monitoring system for detecting falls in the elderly, using recurrent neural networks and low-resolution thermal sensors. While prioritizing privacy and non-intrusiveness, the system achieved a noteworthy 93% accuracy rate in identifying falls. However, certain limitations regarding ambient temperature variations and the presence of objects were identified, indicating the need for further research and enhancements to facilitate practical applications.

The article presented in [17] discusses a fall detection method using an IR-UWB radar sensor combined with a CNN algorithm to address the challenges of privacy preservation, user convenience, and detection performance. The proposed framework collects motion data using a non-contact IR-UWB radar, which is then preprocessed into 2D images suitable for CNN input. The CNN algorithm automatically extracts features from these images to classify behaviors as either “Fall” or “Activities of Daily Living (ADL)”. The study demonstrates the method effectiveness through experiments involving various motions, yielding 96.65% accuracy in distinguishing falls from normal activities.

In reference [22], the authors presented a solution for enhancing safety and independence of elderly individuals in smart home environments. The system uses background subtraction, Kalman filtering, and optical flow to detect falls effectively. Integrated into smart homes, it offers a seamless way to increase both comfort and security. The machine learning aspect processes the inputs from these algorithms, ensuring high detection accuracy. Through testing involving over 50 different fall scenarios, the system achieved a detection rate of over 96%, demonstrating its effectiveness and potential for real-world use.

The work presented in [23] introduces a camera-based fall detection system. This system uses a single top-view depth camera, the Microsoft Kinect, to monitor falls while maintaining the user’s privacy. The detection algorithm works by analyzing depth frames of the human body and training a classifier using a binary support vector machine (SVM) learning algorithm. The application runs on a computer based on Microsoft Windows. Despite its high accuracy rate of 98.6%, this system is not affordable for everyone.

A novel fall detection system leveraging multi-stream 3D CNNs to enhance the accuracy and reliability of fall incident detection is presented in [24]. This vision-based method involves capturing video sequences and using an image fusion technique to preprocess frames into sequences that highlight spatial and temporal differences. These sequences are fed into a four-branch architecture (4S-3DCNN), where each branch processes different fall phases. The model was evaluated using the Le2i fall detection dataset, achieving high-performance metrics. The study demonstrates that the proposed 4S-3DCNN model outperforms several state-of-the-art models, including GoogleNet, SqueezeNet, ResNet18, and DarkNet19, for real-time fall detection in different environments. However, the authors use a computer with high computing requirements to run the approach.

In the literature, many works have excellent metrics regarding accuracy and precision. However, the computational cost of the proposed works is high, directly impacting the system’ s final price. Many systems rely on non-free software, making implementation and maintenance expensive. Thus, a large number of elderly people cannot pay for them. In addition, other systems have technical deficiencies coming from false alerts generated by the presence of objects that may confuse detection. Additionally, lighting issues, environmental variations, recognition of head movements, and even clothing color can contribute to inaccuracies.

## 3. Materials and Methods

This work proposes a non-wearable system for human fall detection. The proposal uses artificial vision to detect falls. Deep learning techniques based on CNN have been implemented for this purpose. The system effectiveness depends largely on the training performed on it. Considering that the proposal is executed on a low-cost SBC, learning transfer techniques were required to optimize the system’s training. The proposal is based on experimental research, which includes:(a)Selection of a lightweight pre-trained model to be easily ported to the embedded system;(b)Data preparation which involves creating or curating datasets to be used for training and validating the deep learning model;(c)Adjustment of the model parameters to find an adequate response for the problem;(d)Model training on large and diverse datasets and tuning hyperparameters;(e)System integration on the SBC platform;(f)Performance evaluation by means of metrics such as accuracy, sensitivity, precision, specificity, as well as ROC AUC scores for defining the threshold value.

The methodology used for the implementation of the proposal is divided into two main stages:The training stage;The processing stage.

A computer provided with a graphics processing unit (GPU) is required to perform fast and efficient training during the first stage. Once the pre-trained model is obtained, it is evaluated in real scenario by the processing stage. If the system response is not satisfactory, the neural network training stage must be repeated to improve the accuracy and sensitivity of the system. The methodology is summarized in the block diagram shown in Figure 1.

### 3.1. Model Selection

CNNs have proven to be very effective in the field of computer vision [25,26]. Therefore, this work is based on this architecture to extract features and classify image objects. The classification is done based on the probabilistic distribution of the detected object delivered by the network. Subsequently, if the system predicts a fallen person with a probability above a set threshold, a notification will be sent via text message. This proposal classifies detected objects into the following 5 classes:Fallen person;Crouching person;Sitting person;Standing person;Person lying down.

This work uses the combined model SSD-MobileNet-v2. It consists of two main modules: MobileNet-v2 as a feature extractor and Single Shot Detection (SSD) as an object locator. The MobileNet-v2 module is suitable for its compact size, acceptable accuracy, and prediction speed. MobileNet-v2, compared to MobileNet, introduces two new features: linear bottlenecks and shortcut connection possibility between input/output bottlenecks [27].

The MobileNet-v2 is conformed by convolutional blocks, where each block is composed of three stages. The first stage performs a 1 × 1 (pointwise) convolution with the Rectified Linear Unit (ReLU) function as a non-linearity function [28]. The second and third stages compute a depthwise separable convolution. In the second stage, a depthwise convolution filters the input channel. For that, it splits height × width dimension (*h* × *w*) from depth dimension (*k*), with *k* being the number of the input channels. The depth of the output depends on the expansion factor (*t*) given by the number of characteristics extracted by the layer. Finally, a new feature is obtained in the third stage by computing the depthwise convolution (dwise) result with a 1 × 1 convolution. The use of depthwise separable convolution reduces computation and model size [29]. MovileNet-v2 block bottleneck model is summarized in Table 1.

The architecture of MobileNet-v2 model is shown in Table 2, where *t* is the expansion factor, *c* is the number of output channels, *n* is the repeating number, and *s* is the stride. In the proposed architecture, the first layer is a convolution layer with an input of a 320×320×3 image, which means an RGB image with dimension of 320×320 pixels. For the first layer, each input channel corresponds to one color channel (red, green, or blue). It is important to note that the other depthwise convolution layers of the model use 3 × 3 kernels for spatial convolution.

The SSD model performs object detection and generates multiple bounding boxes for the defined object classes. In this way, the model can detect multiple objects by applying convolutional filters to default bounding boxes [31]. To estimate the detection, it evaluates the default boxes with different aspects with feature maps having different scales. For each box, it predicts confidence values for all object categories. Then, each detected object class instance is estimated with a probability value. For the prediction task, convolutional filters are used. The kernel element used is a 3×3×ci, where ci is the number of channels [32]. The architecture of SSD model is shown in Table 3. The complete SSD MobileNet-v2 model is summarized in Figure 2.

### 3.2. Pre-Trained Model Selection

The training of the network was performed using the Tensorflow framework. In order to optimize resources, the pre-trained model ssd_mobilenet_v2_fpnlite_320 × 320_coco17_tpu-8 was selected [33]. This model was trained with the COCO (common objects in context) dataset for the classification of 91 object types, with “person” being one of the super classes [34]. This pre-training is useful because the model has learned to recognize persons based on a large dataset of more than 2 million images from the COCO dataset.

To recognize the five predefined poses of the person (standing, crouching, sitting, lying down, and falling), the last layers of the model were replaced. Therefore, it was necessary to define and prepare the dataset of images for classification. For this purpose, open-access datasets were used. To improve the sensibility, a combined dataset was generated by by combining the UR Fall Detection dataset and our own dataset. This new combined dataset allows training the network in a more efficient way and obtaining the trained model.

### 3.3. Dataset Preparation

On the ond hand, the UR Fall Detection dataset, presented in [35], consists of 8065 images, divided into 70 image sequences representing both daily activities and falls (30 image sequences). On the other hand, our dataset consists of 19,584 images given a total of 27,649 images. They were captured using a Python application, normalized to a 640 × 480 × 3 dimension, and prepared using the LabelImg, which is an open source annotation tool. This tool allows user to select the region of interest from the image that contains the object and label it with the respective predefined class. The LabelImg then generates a xml file for each image with the corresponding labeled information.

Selecting the appropriate region of interest during dataset preparation is crucial to ensure accurate predictions, especially when distinguishing similar classes, such as falling and lying down. If the region of interest is too close to the person, it may omit useful contextual information, leading to inaccurate predictions. Conversely, including adequate contextual information in the region of interest can enhance the model’ s prediction.

During training and validation processes, it is necessary the original image with it corresponding labelling file (.xml). Prior to train and validate, two binary TFRecords files were generated, one for the training process and the other for the proposal’ s validation. The training set consisted of 80% of the images (22,120), while the remaining 20% corresponded to the validation set (5529).

### 3.4. Network Training

During the CNN training process, the backpropagation algorithm was used. Through iterative and recursive operations, this method adjusts the weight of the CNN nodes to minimize the gradient of the error function and improve the predictive model capacity. In order to facilitate the learning process and introduce non-linear behaviors to the network, the ReLU and Softmax activation functions were used. ReLU was used to learn complex patterns with a low computational cost, and Softmax or normalized exponential function, to perform the multi-class classification of the objects.

The network’ s training was performed using the Tensorflow framework, and Tensorboard was used as a neural-network analysis tool to visualize the process. This evaluation was performed with Tensorboard on an AMD Rizen 7 laptop (By Hewlett-Packard in Palo Alto, CA, USA), with Radeon Graphics 2.90 GHz and 12 GB of RAM, equipped with an NVIDIA GeForce GTX 1650 Ti GPU (By NVIDIA Corporation, Santa Clara, CA, USA) with 4 GB of dedicated video memory to speed up the training process. The training dataset was divided in batch size of 4 images, then 5530 iterations completes an epoch. For the validation of the model, 20% of the prepared dataset was used. It is important to note that the validation set was build with images that contain different people and objects from those used during the learning process. This can be appreciated in the total loss training and validation curves, both illustrated in Figure 3. The x axis in the curves represents the number of iterations. It can be observed that at fourth epoch, it seems to begin an overfitting; however, after continued training, it disappears and both curves tend to converge. The total loss curves are illustrated for 18 epochs.

The total loss function considers classification, localization, and regularization losses. For classification, weighted sigmoid focal loss function was used; while for localization, weighted smooth L1 loss was used. Training results showed a total loss function value of 0.069 for epoch 18th and a learning rate of 0.079. These values were considered acceptable for starting the functional tests in real conditions. Once the model was validated, learning was transferred to the target device used for fall detection. The trained model has a weight of 19.8 MB, which guarantees that it can be embedded inside an application executed on a SBC.

### 3.5. Processing Stage

In this stage, the model was evaluated under real conditions, and results were used to fine-tune the system. The embedded application was executed on a Raspberry Pi4 with 4 GB RAM. The images were captured by an HD mechanical IR-CUT camera. The block diagram of the application is shown in Figure 4. The application was developed in Python programming language, and the OpenCV library was used to acquire and process the image in real time [36].

The flowchart of the application is shown in Figure 5. The program initializes the pre-trained model, captures the image, converts it into an input tensor for the network, and then executes the analysis of each frame, applying the neural network model to predict falls. The Mobilenet-v2 layers are used to obtain the features whereas the SSD multi-locator identifies if an object of a given class is present in the image. If the model recognizes objects of one of the five predicted classes in the image, boxes are drawn on the image with probability values. Depending on threshold, the occurrence of the fall is predicted. Lastly, the algorithm sends an alert notification to a mobile device, including a picture of the event. This was performed via the Telegram instant messaging application through a bot, so more than one user can receive the alert. For that, the system requires an Internet connection. Note that the system uses the Minimum Bayes Risk criterion to decide whether to send the alert notification, since the most concerned issue is the detection of a Fallen person. For example, if two classes are predicted at the same time, system prioritizes Fallen Person regardless of whether the other class has a higher probability, since the goal is to reduce the more expensive outcome.

### 3.6. Verification Tests

Verification tests were performed in two stages. The first one consists of a functional test and the second one was conducted in a controlled environment to measure the performance metrics of the system. The functional test was first performed to verify the detection of the five classes of objects detected by the system, as shown in Figure 6. Through the functional tests, it was also checked that the system recognizes more than one object.

Furthermore, various poses of fallen persons were tested to analyze the morphological system response. Figure 7 illustrates the response of the fall detection system. The tests were set up with the camera positioned at 1.5 m height.

In addition, functional tests are useful to verify whether the Labeling process had included adequate contextual information in the region of interest, to distinguish between similar classes such as Person lying down and Fallen person. Prediction results in living room and bedroom are illustrated in Figure 8 and Figure 9. In both figures, the top images are related to labeling examples of our dataset, whereas the bottom images are related to predictions.

During the second stage, for evaluating the system, people activities were divided into ”Fall” and ”Other Activities”. The verification tests were performed considering the confusion matrix. The four possible outcomes are:True positive (TP): The person falls and the system predicts a fallen person.True negative (TN): The person does other activities and the system predicts other activities.False positive (FP): The person does other activities and the system predicts a fallen person.False negative (FN): The person falls and the system predicts other activities.

Three different scenarios with controlled environment were studied to obtain system performance metrics. To evaluate the capacity of the system to distinguish falls from other activities, three variables of interest were considered: outfit, light level, and distance to object. Tests take into account three different conditions for each variable. For example, light level values were set considering general indoor activities (gloomy 15 lux, satisfactory 100 lux, and good 250 lux). This was done because the system focuses on nursing homes.

In this experiment, 14 participants aged between 26 and 84 were recruited. Each participant completed the test design shown in Figure 10, which include 27 possible combinations for the three variables of interest. The test design was executed twice: the first one to test fall detection and the second one to test other activities detection. Thus, 54 tests were performed for each participant, resulting in 756 tests. Tests were performed with different scenarios to obtain the overall system analysis.

The performance indicators used for analyzing the results are: accuracy, precision, sensitivity, and specificity, presented in Equations (Equation 1)–(Equation 4). Accuracy represents the system’s ability to correctly distinguish a fall from other activities. Precision represents the number of fall alerts that are actual falls. Sensitivity indicates how capable the system is to detect an actual fall as a fall. Finally, specificity indicates the capability of the system to detect actual other activities as other activities:(1)$$Accuracy=\frac{TP+TN}{TP+TN+FP+FN}\times 100$$
(2)$$Precision=\frac{TP}{TP+FP}\times 100$$
(3)$$Sensitivity=\frac{TP}{TP+FN}\times 100$$
(4)$$Specificity=\frac{TN}{TN+FP}\times 100$$

## 4. Results

This section presents the behavior of the fall detection system depending on the variable of interest. The first parameter under study is the outfit. From the results depicted in Figure 11, it can be seen that the system performs better when individuals are dressed in ordinary or fluorescent outfits, whereas performance decreases when participants are wearing black/white (B/W) clothing. In fact, it was observed that dark clothing affected the detection.

The next parameter to be considered is the light level of the environment. From the results shown in Figure 12, it can be observed that the system performs better under higher light levels, whereas fall detection becomes slightly more difficult in gloomy environments. Note that the system is intended to work in bedrooms or living rooms where the light levels range from 100 to 200 lux.

The last variable to analyze is the distance from the camera to participant, as observed in Figure 13. In this case, the system tends to experience a greater number of failures when the person is distant from the camera, whereas performance increases as the participant gets closer to the camera. False Negatives (FN) considerably increase in distances greater than 3 m. From experimental data, it can be established that 11 of the 21 FN were predicted at a low light level (15 lux), 8 FN at 100 lux, and 2 FN at 250 lux. Regarding the 5 FN predicted in other distance conditions, all were performed under low light level. Hence, it can be concluded that two conditions mainly affect the response of the system: distance from camera over 3 m and low light level (15 lux). The distance fact can be explained because the neural network has not been enough trained with a dataset considering larger distances, since rooms are commonly less than 4 m long.

Figure 14 summarizes the global obtained results of the research. It can be seen that the system prediction is good since it provides 352 true positives (fall detection) and 365 true negatives (other activities detection). Also, the confusion matrix results allow obtaining the system performance, both for specific and global analysis.

Figure 15 illustrates the Receiving Operating Characteristic (ROC) curve for the experimental test, which represents the performance of our binary classifier (falls and other activities) in real conditions. It can be seen that the system has a good performance having an AUC of 0.957, which means a good prediction. The inflection point of the curve (0.03439153, 0.93121693) defines a 0.4 threshold value. At this point, a sensitivity (FPR) of 93.1% was obtained. Even though the FN doubles the FP as observed in the confusion matrix, this point gives the best prediction. It is important to note that 81% of FN occurred at distances over 3 m.

Recalling that one in three adults over the age of 65 experiences a fall annually, and given that our system predicts 93 of a hundred falls, the probability of not detecting a fall annually is about 2.33%, with 1.88% not being predicted at distances over 3 m.

The calculated metrics of the system are summarized in Table 4. Based on these values, it can be said that the system’s performance is good enough for many applications, such as care in nursing homes.

## 5. Discussion

This study is based on computer vision where a camera captures images and then sent them to a CNN responsible for analyzing data and providing the probability of having a fallen person. Since our system stands on a SBC, it is a cost-effective and high performance solution, costing around 150 USD. Table 5 compares our system with similar ones, considering relevant aspects such as portability, hardware and software components, and performance metrics.

Compared to other studies that rely on specific devices such as motion sensors or accelerometers [8,9,10,13,14], this proposal presents a non-intrusive alternative for detecting falls. It eliminates the need to carry additional devices resulting in more acceptance and comfort for users. It also has to be considered that wearable solutions have a disadvantage, i.e., the use of batteries that have to be recharged or replaced after a certain period of time. However, current manufacturing technology allows for minimizing energy consumption extending its autonomy. Another advantage of our approach is that when a fall is detected, the system sends an image via a Telegram message to notify the event, which enables visual confirmation, thereby reducing false alarms.

The global metrics presented in this work are: accuracy of 94.8%, sensitivity of 93.1%, precision of 96.4%, and specificity of 96.6%. Accuracy and sensitivity metrics indicate that the prediction of falls and other activities are adequate for the intended use. Current metrics certainly could increase in bedrooms or living rooms with satisfactory light levels (e.g., >100 lux). In case of total darkness, the performance metrics remain consistent due to the camera being equipped with IR LEDs. Concerning distance to the camera, accuracy and sensitivity can be improved by increasing our image dataset, especially when the distance is more than 3 m from the camera. On the other hand, the precision value indicates that there is a low probability of having false alarms, which is complemented by the confirmation provided by the picture sent by the system.

Compared to [19], which implements a multi-camera system for covering large areas with 89% accuracy, our proposal, which employs a lightweight CNN implemented on a SBCs with limited computational resources, achieves a higher accuracy with a difference of 5.8%. Additionally, our experimental validation was conducted with individuals under various conditions of clothing, distance from the camera, and lighting, allowing for a more comprehensive evaluation of the system.

The authors of [17] presented a system based on IR-UWB radar which ensures the preservation of privacy. The system is capable of classifying falls versus common daily activities. However, it does not have the ability to differentiate between a person lying down and a fallen person, unlike our system. In terms of performance metrics, their work have a bit better accuracy (1.85%), whereas our proposal has a bit better sensitivity (0.9%), despite both systems employing a CNN implemented in different hardware.

In the work presented in [21], a non-wearable system is implemented using channel state information (CSI), which detects variations in WiFi signals within a physical environment. Comparing both approaches, their system exhibits a slightly better accuracy and sensitivity with a similar precision. The work presented in [1] is a system based on thermal sensors employed to detect the body heat of the individual. Similar to our approach, sensor data processing is carried out on a SBC. In terms of performance, our system shows slightly better accuracy, sensitivity, and specificity.

In [23], a fall detection system based on a Kinect sensor configured in a top-view setup is presented. It preserves privacy by using depth data instead of detailed images. The system employs an SVM algorithm and demonstrates high accuracy. However, it was tested in a small area of 8.25 m² and it encountered difficulties in detecting fallen individuals due to noise present in the depth frames.

An outstanding work is presented in Alanazi et al. [24]. It describes a vision fall detection method using a 3D-CNN with metrics around 99%. However, to obtain those results, authors use a computer with an excellent performance (Intel Core I9-9900K CPU 3.6GHz, 64GB RAM with NVIDIA GeForce RTX 2080Ti GPU (By NVIDIA Corporation, USA)) compared to our embedded system (Raspberry PI 4, Quad core ARM Cortex A72 CPU @ 1.8 GHz, 4 GB RAM (By Sony in Pencoed, Gales)). Nevertheless, a better computer would significantly increase the costs of the system. Furthermore, it is important to note that our results are based on experiments with real persons instead of using videos or photos for testing stage. In addition, our proposal allows the effective detection of several persons simultaneously, which is a benefit compared to wearable devices.

Concerning privacy, even though our system uses a camera, video is not recorded, and pictures are only sent when a fall is detected. If the system has multiple cameras positioned strategically in the room, as in [37], the performance certainly could increase. However, users might feel a greater invasion of their privacy.

## 6. Conclusions

This work proposes a fall detection system based on computer vision and Convolutional Neural Networks. The lightweight SSD-MobileNet-V2 model architecture was implemented to be capable of running on a Single Board Computer. For the training process, a mixed dataset combining UR Fall Detection (8065 images) with our own dataset (19,584 images) was used.

Results exhibit satisfactory performance (AUC = 0.957) considering that this is an affordable solution to be implemented in nursing or elderly homes. An efficient alert system was designed, allowing caregivers and relatives to receive an image notification. Although there are some inexpensive commercial solutions, they involve a monthly fee for the service. Moreover, it works as long as there is a medical alert service in the country.

Several tests were conducted to determine system performance under different lighting conditions, clothing, and distance from the camera. The system achieves its best performance from an intermediate light level onwards. Regarding clothing, there is a slightly reduction on system sensitivity when the person wears dark clothing.

In future research, the system will be enhanced by extending the dataset with low-light level images and increasing the distance between the person and the camera in order to improve system sensitivity. In addition, other occlusion techniques such as image inpainting would be tested. In order to preserve individual privacy in spaces such as bathrooms, alternatives like detecting abrupt changes in Wi-Fi signal patterns through CSI can be explored for fall detection. Additionally, incorporating this method will provide dual detection capabilities, thereby minimizing the likelihood of false negatives.

## Figures and Tables

**Figure 1 sensors-24-05592-f001:**
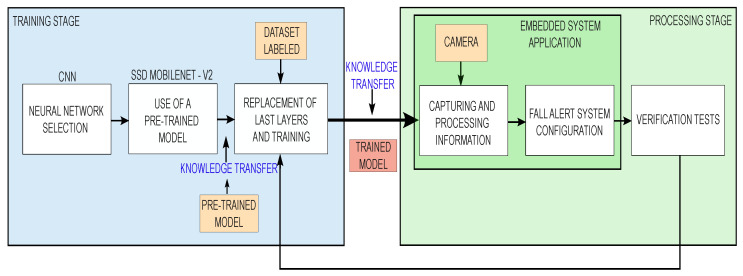
Methodology.

**Figure 2 sensors-24-05592-f002:**
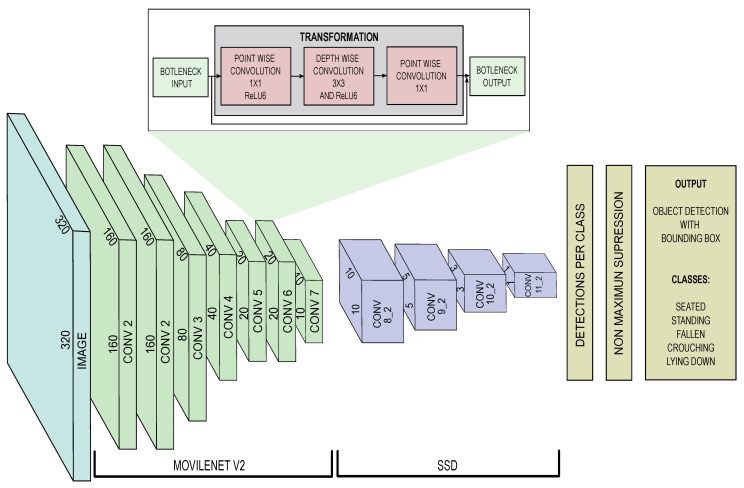
SSD Mobilenetv2 architecture.

**Figure 3 sensors-24-05592-f003:**
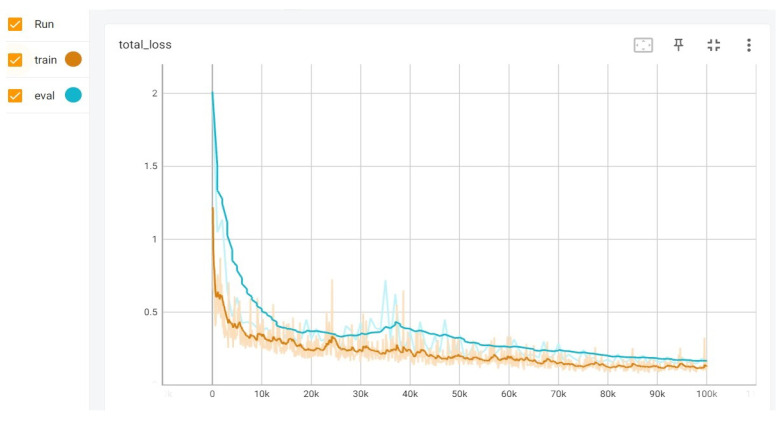
Total loss curves.

**Figure 4 sensors-24-05592-f004:**
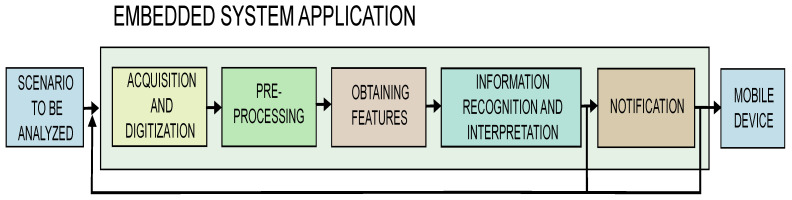
Embedded application block diagram.

**Figure 5 sensors-24-05592-f005:**
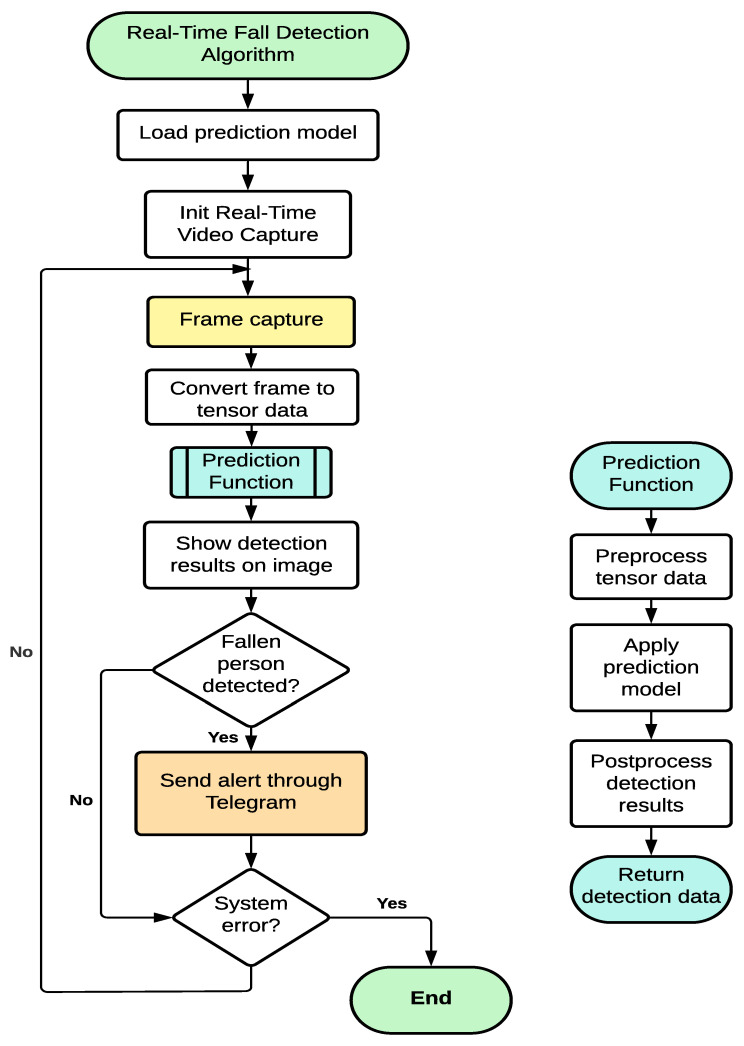
System flowchart.

**Figure 6 sensors-24-05592-f006:**
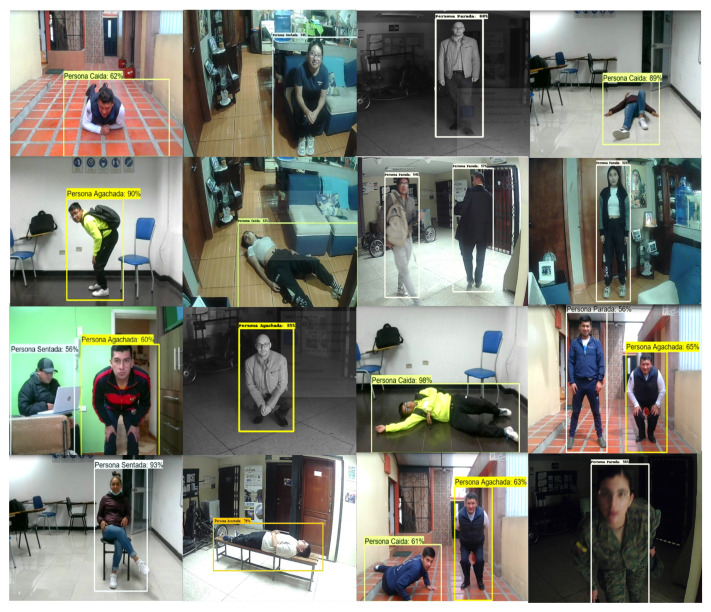
Classes detected: Fallen person (“Persona caída”), Crouching person (“Persona agachada”), Sitting person (“Persona sentada”), Standing person (“Persona parada”), Person lying down (“Persona acostada”).

**Figure 7 sensors-24-05592-f007:**
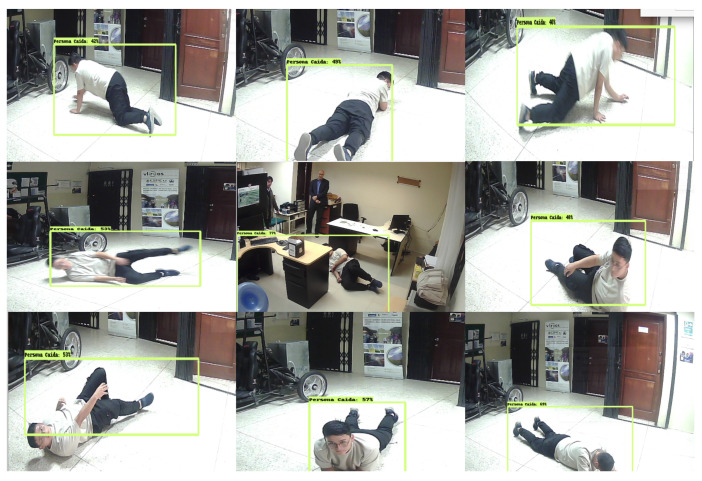
Different poses of a “Fallen person”.

**Figure 8 sensors-24-05592-f008:**
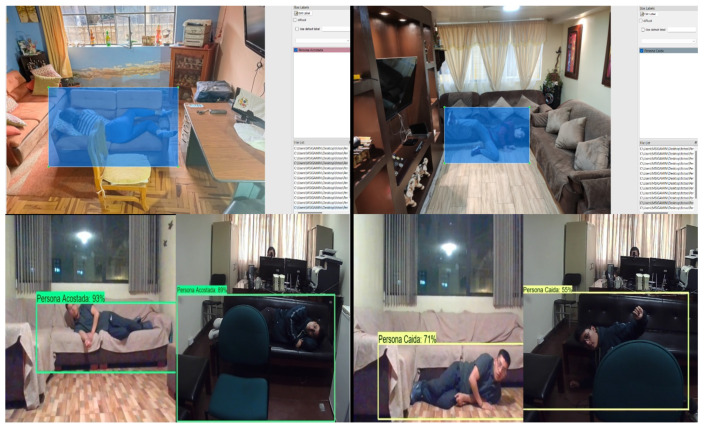
Prediction results in living room considering contextual information.

**Figure 9 sensors-24-05592-f009:**
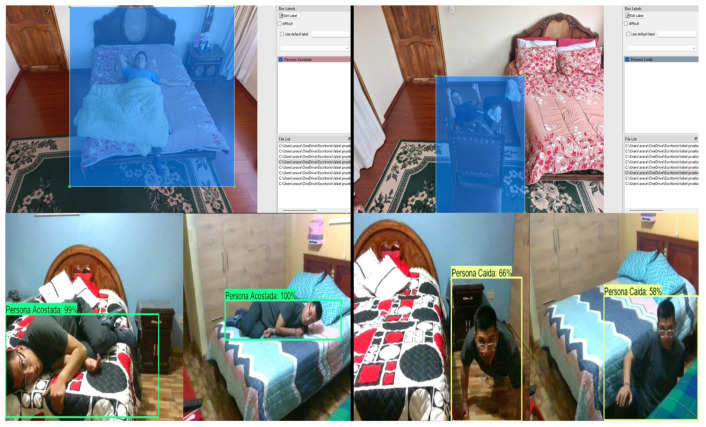
Prediction results in bedroom considering contextual information.

**Figure 10 sensors-24-05592-f010:**
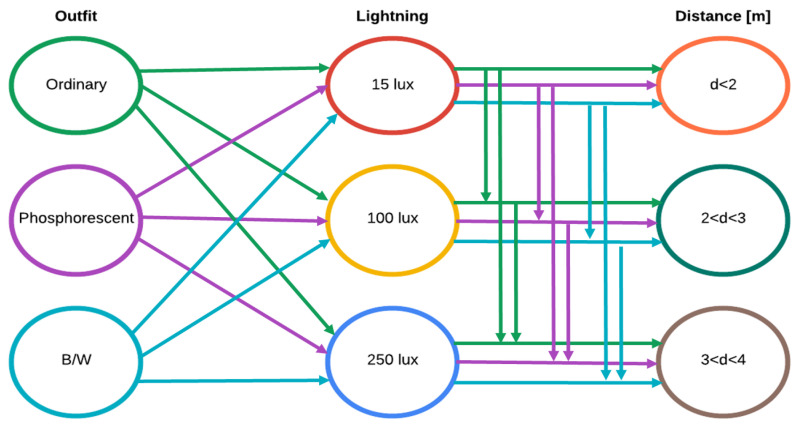
Test design under different outfit, lightning, and distance conditions.

**Figure 11 sensors-24-05592-f011:**
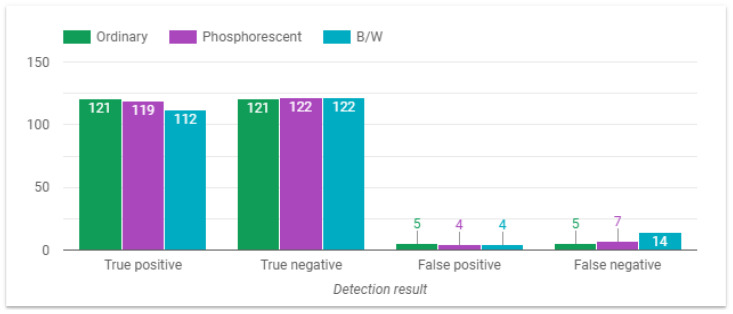
System performance with different outfits.

**Figure 12 sensors-24-05592-f012:**
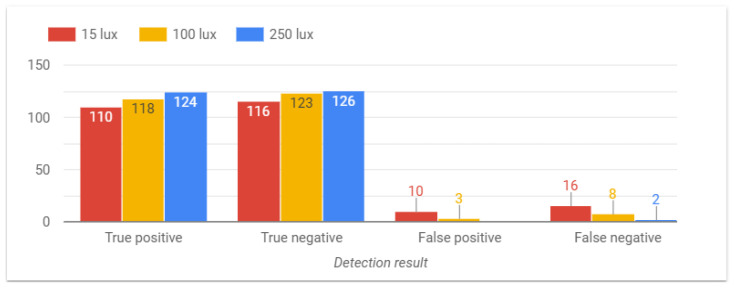
System performance under different lightning levels.

**Figure 13 sensors-24-05592-f013:**
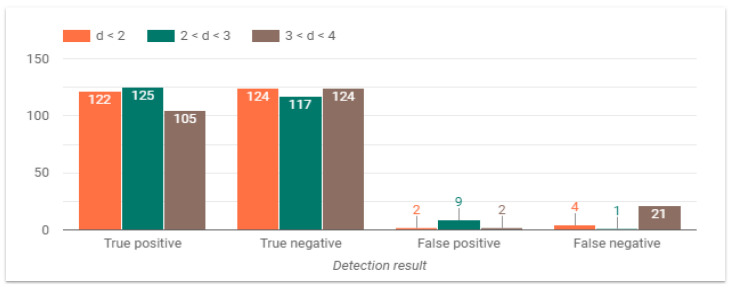
System performance at different distances from the camera.

**Figure 14 sensors-24-05592-f014:**
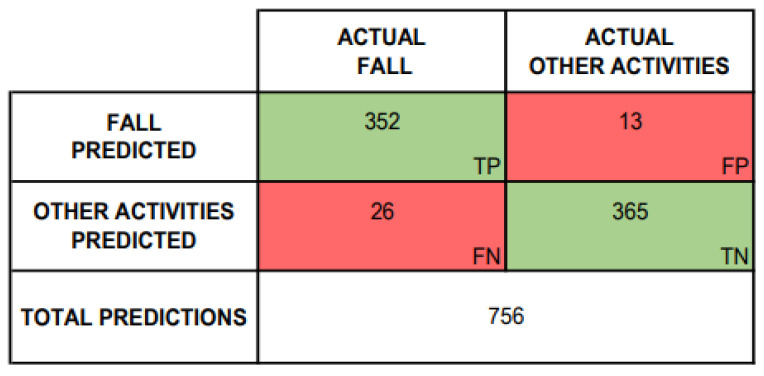
Confusion matrix of tested scenarios.

**Figure 15 sensors-24-05592-f015:**
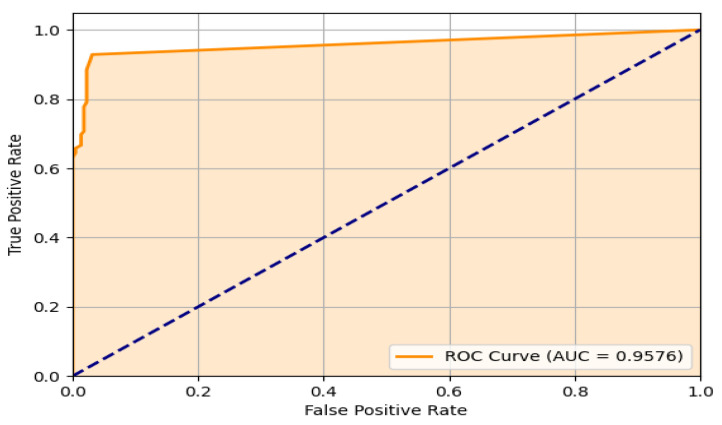
ROC curve of the proposed system.

**Table 1 sensors-24-05592-t001:** Stages of the MobileNet-v2 bottleneck residual block [30].

Stage	Input	Operator	Output
1st	h×w×k	1×1 conv2d, ReLU6	h×w×(tk)
2nd	h×w×tk	3×3 dwise s = *s*, ReLU6	$\frac{h}{s}\times \frac{w}{s}\times \left(tk\right)$
3rd	$\frac{h}{s}\times \frac{w}{s}\times tk$	Linear 1×1 conv2d	$\frac{h}{s}\times \frac{w}{s}\times k^{\prime }$

**Table 2 sensors-24-05592-t002:** MobileNet-v2 layer details.

Name	Type	Input (*h* × *w* × *k*)	Operator	*t*	*c*	*n*	*s*	Output
Conv 1	Convolution	320×320×3	Conv2d	-	32	1	2	160×160×32
Conv 2	Depthwise Conv	160×160×32	Bottleneck	1	16	1	1	160×160×16
Conv 3	Depthwise Conv	160×160×16	Bottleneck	6	24	2	2	80×80×24
Conv 4	Depthwise Conv	80×80×24	Bottleneck	6	32	3	3	40×40×32
Conv 5	Depthwise Conv	40×40×32	Bottleneck	6	64	4	2	20×20×64
Conv 6	Depthwise Conv	20×20×64	Bottleneck	6	96	3	1	20×20×96
Conv 7	Depthwise Conv	20×20×96	Bottleneck	6	160	3	2	10×10×160

**Table 3 sensors-24-05592-t003:** SSD layer details.

Layer	Type	Grid Size	Kernel Size (h × w × ci)	Output
Conv 8_2	Convolution	10×10	1×1×256 3×3×512−s2	10×10×512
Conv 9_2	Convolution	5×5	1×1×128 3×3×512−s2	5×5×256
Conv 10_2	Convolution	3×3	1×1×128 3×3×256−s1	3×3×256
Conv 11_2	Convolution	1×1	1×1×128 3×3×256−s1	1×1×256

**Table 4 sensors-24-05592-t004:** Global system performance analysis.

Variable	Condition	Accuracy [%]	Precision [%]	Sensitivity [%]	Specificity [%]
	Ordinary	96	96	96	96
Outfit	Phosphorescent	95.6	96.7	94.4	96.8
	B/W	92.9	96.6	88.9	96.8
	15 lux	89.7	91.7	87.3	92.1
Illumination	100 lux	95.6	97.5	93.7	97.6
	250 lux	99.2	100	98.4	100
	d < 2	97.6	98.4	96.8	98.4
Distance (d)	2 < d < 3	96	93.3	99.2	92.9
	3 < d < 4	90.9	98.1	83.3	98.4
Overall performance		94.8	96.4	93.1	96.6

**Table 5 sensors-24-05592-t005:** Comparison table with other studies.

Reference	Method Portability		Hardware Devices				Software Machine Leanring		Fall Detection Results Metrics			
	Wearable	Non Wearable	Sensors	Camera	Alert Service	Controller/CPU	Algorithms	Dataset	Accuracy	Sensitivity	Specificity	Precision
Yacchirema et al. [8]	X	-	LSM6DS0 (acc+gyro)	-	MQTT Brocker	Smart IoT Gateway (RPI3+STM32)	Ensemble	SisFall	98.72%	94.60%	99.48%	96.22%
Vallabh et al. [10]	X	-	Cellphone (acc+gyro)	-	-	CPU N/S	k-NN ANN SVM	MobiFall	k-NN: 87.5% ANN: 85.47% SVM: 86.75%	k-NN: 90.7% ANN: 89.23% SVM: 89.74%	k-NN: 83.8% ANN: 81.43% SVM: 82.93%	-
Badgujar & Pillai [9]	X	-	ADXL345 (acc)	-	-	CPU N/S	Decision trees. SVM	SisFall	Decision trees: 95.87% SVM: 84.17%	-	-	-
Torti et al. [13]	X	-	LSM6DSM (acc+gyro)	-	-	SensorTile	RNN: LSTM	SisFall	98.33%	98.73%	97.93%	-
Perejón et al. [14]	X	-	ADXL345 (acc)	-	N/S	STM32	RNN: GRU	SisFall	96.70%	87.50%	96.8%	68.1%
Shu & Shu [19]	-	X	-	X	Telegram	Eight-core Amlogic S912 cortex-A53 CPU	SpeedyAI RVM	SpeedyAI images. Videos of authors.	89%	-	-	-
Ricciuti et al [23]	-	X	Microsoft Kinect v1	-	-	CPU N/S	SVM	Own Dataset	98.6%	-	-	-
Han et al. [17]		X	IR-UWB radar	-	-	CPU N/S	CNN	Own Dataset	96.65%	92.2%	-	88.18%
Taramasco et al. [1]		X	OMRON D6T-8L-06 (temp)	-	Own application	ODROID-C1+	RNN: LSTM GRU Bi-LSTM	Own Dataset	LSTM: 91% GRU: 87.5% Bi-LSTM: 93%	LSTM: 89% GRU: 85% Bi-LSTM: 93%	LSTM: 93% GRU: 89% Bi-LSTM: 93%	-
Chu et al. [21]		X	INTEL 5300 NIC (for CSI record)	-	-	Intel NUC D34010WYH	CNN: EfficientNet	Own Dataset	96.8%	96.9%	-	96%
Alanazi et al. [24]		X	-	X	-	Intel Core i9-9900K 3.60 GHz	3D CNN	Le2i Fall Detection	99.03%	99%	99.68%	99%
Our work	-	X	-	X	Telegram Bot	Raspberry Pi 4B	CNN	UR Fall Detection Own Dataset	94.8%	93.1%	96.6%	96.4%

N/S: Not specified. -: Not reported.

## Data Availability

The raw data supporting the conclusions of this article will be made available by the authors on request.

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
