# Peer review of "Low-Cost Non-Wearable Fall Detection System Implemented on a Single Board Computer for People in Need of Care"

_sensors, 2024, doi:10.3390/s24175592_

Round 1

Reviewer 1 Report

Comments and Suggestions for Authors

The article "Low-Cost Non-Wearable Fall Detection System Implemented on a Single Board Computer for People in Need of Care" submitted by the authors for publication addresses a specific theme of transformations in social life on an advanced technological support, intelligent life and stupid death. The article, by way of writing, refers to the socio-economic context of the authors' country of origin, thus suggesting that in that context their proposal, their investigation is performing well, taking into account the lack of resources. I recommend the authors to avoid this confusion of science and everyday life in this article and in the future. Of course, many technologies are available today at a lower price than the one proposed in the article as Smart Phone applications or extensions. So from this perspective, your solution has a price/quality ratio below the market. The experimental setup, image processing, algorithms are correctly presented, but nothing new. A scientific contribution is needed here, the internet is full of such realizations, even more complicated in the IoT, hobby area. Many examples, many software libraries are already available, so I can only congratulate you for learning to use them and I wish you original ideas and a scientific contribution outside the social context.

The article is well written, but no personal contribution is highlighted to demonstrate that more can be done with fewer resources in another way. Unfortunately, critical remarks are made in the article about the more sophisticated equipment that contains adequate sensors. Yes, the image is important, image processing is important, but without those useless sensors in your concept, we do not see contextual information and we process pixels, not real information. If a man ties his shoelaces, the medical services will come, it is a superficial approach, but a useful programming exercise for you.

Reviewer 2 Report

Comments and Suggestions for Authors

In this article an activity classification method is explained. It is mainly used for fall detection.

I have some comments:

1) Detection of falls in older people is an active research topic. The authors defend that their system is cheap. But it doesn't mean it's the cheapest. It is not a well-explained advantage since, for example, there are detection systems that use a simple accelerometer. Obviously, an accelerometer is cheaper than a set of cameras.

2) Fall detection using cameras also has the drawback of being limited to some areas of the house. And it is not possible to use it outdoors.

3) Privacy is another problem. It can be guaranteed that the images are not used for purposes other than detection. But the main problem, in my opinion, is feeling observed. Let's think for example, a person in the bathroom. It is a place where more falls tend to occur. A camera watching a person taking a shower is not ideal.

4) In relation to the developed method, I understand that it seeks to classify the user's activity associated with the closest objects. Perhaps the drawback of this is that if the user performs a task (+object) not included in supervised learning, it is not known what the prediction model will decide. This is why it is not clear that a camera system has fewer false positives than, for example, an accelerometer.

5) In relation to the previous comment, the classification system distinguishes between lying down and falling. In my opinion, explainability techniques should be applied to know in which parts of the image the prediction model focuses its attention between these two situations. For example using occlusion techniques.

6) Why hasn't a simple binary classification (fall, not fall) been made?

7) I think learning and validation curves should be shown (error vs epochs). These curves should be obtained in such a way that users and objects that have participated in the learning process do not participate in the validation. This way you can see if the learning is enough to avoid overfitting and if many more images are needed for training.

Reviewer 3 Report

Comments and Suggestions for Authors

The paper describes the experimental research devoted to an emerging topic related to injury prevention. The paper contains all necessary parts of the research article and generally written quite well. The strong point of the paper is its experimental base.

General remarks

1.     As the classification problem is involved, it is crucial to define the criterion which is used. Such a criterion can be Neyman-Pearson, Minimum Bayes Risk (Unweighted or Weighted) or something else. This choice is determined by the nature of the primary problem to be solved. In case of the system proposed in the paper, the most ‘expensive’ outcome (in terms of weights or penalties applied to the confusion matrix entries) is FN as it is described in line 266. This is the outcome where the person fell and the system missed it and nobody was informed. We must avoid this at all cost! The opposite outcome (FP) is significantly less ‘expensive’, since the system will send a picture to a person in charge and there is another pair of human eyes to check and take up the final decision about the outcome. The confusion matrix in fig. 11 does not reflect this natural conjecture, where FN is twice as many as FP.

2.     The confusion matrix shown in figure 11 does not seem to be sufficient for the evaluation of the classifier. Since the authors deals with binary classification problem, the ROC-curve (with its metric: AUC or AOD) is essential because it is the commonly-used way to describe the performance of any classifier. Being properly scaled, ROC can also show the possibility of the trade-off between TN and FP.

3.     The loss function that was chosen for CNN training has not been described explicitly in the paper. It makes difficult to interpret its values which are given in line 224.

Specific remarks

1.     In line 27, it is not clear how gender can be a factor determining the severity of the fall consequences.

2.     In line 177, the clarification is necessary for the depth dimension of the input data. Is it the color components RGB or something else?

3.     In figure 4, it is not clear how the rhombus node with ‘True’ inside should work. In algorithm flowcharts, such a joint point is typically drawn just as an arrow connected to the main flow line.

4.     In figure 4, it is not obvious how statement ‘q key pressed’ can be checked. If this is algorithm to be executed on SBC, who and how can press this button?

5.     After examining figures 7-10, I would recommend to paint each node of the chart in figure 7 with the color which is used in the bars in figures 8-10.

6.     In figure 9, the order of the bars does not seem to be natural. It should be either increasing (15, 100, 200 lux) or decreasing (200, 100, 15 lux).

7.     The statement in line 331 is ambiguous: ‘In case of total darkness, metrics remain the same since the camera is equipped with IR LEDs.’ If it means that the performance is the same (the values of the metrics are the same), then this statement has no proof in the paper. Moreover, the reviewer supposes that the performance will be poorer than it was in B/W scenario due to the nature of IR images.

Summary

The paper can generally be recommended for publication after the remarks are properly addressed. The reviewer supposes that the amount of raw data acquired in the series of the conducted experiments should allow the authors to improve the statistical verification of their system.

Comments on the Quality of English Language

The quality of English is quite fine, only few mistakes can be found. For instance, in line 27, ‘for elderly individuals’ should probably be instead of ‘in elderly individuals’.

Round 2

Reviewer 1 Report

Comments and Suggestions for Authors

The authors of the article "Low-Cost Non-Wearable Fall Detection System Implemented on a Single Board Computer for People in Need of Care" rewrote some sections and improved the first version during the review. The article is not yet in the final stage, out of zeal they repeated the new elements from the introduction section: a=b, d=g. It doesn't matter how many new elements there are and they should not be artificially increased, the important thing is that they are real. 

The argument that a subscription is not necessary is not related to the scientific side but to the commercial policy of a company operating a product - it has nothing to do with science or the value of the article.

Reviewer 2 Report

Comments and Suggestions for Authors

Thanks for your answers.

I would have liked to see the result of using, for example, occlusion techniques on falling and lying down images. It would have been interesting to see which part of the image the prediction model focuses on to make a prediction between those two cases.

On the other hand, I think that changing a camera in the visible domain for a thermal one does not reduce the rejection of being observed.
